# Driving Factors and Trend Prediction for Annual Sediment Transport in the Upper and Middle Reaches of the Yellow River from 2001 to 2020

**Jingjing Wu, Jia Tian ***, **Jie Liu, Xuejuan Feng, Yingxuan Wang, Qian Ya and Zishuo Li**

School of Agriculture, Ningxia University, Yinchuan 750021, China
*  Correspondence: tianjia@nxu.edu.cn

**Abstract:** The Yellow River has long been known for having low water and abundant sediment. The amount of sediment transported in the upper and middle reaches of the Yellow River (UMRYR) has changed significantly in recent years, resulting in an obvious imbalance in the spatiotemporal distribution of the water resources in the Yellow River Basin (YRB). The changes in the sediment transport in the Yellow River significantly affect ecological security and socioeconomic development in the YRB. In this study, the Google Earth Engine (GEE) platform was used to obtain the potential driving factors influencing the five main gauge stations in the UMRYR: vegetation, soil moisture, population, precipitation, land types, etc. The data on the annual sediment transport (AST) were from the River Sediment Bulletin of China (2001~2020). Linear regression and the Mann–Kendall test were used to study the temporal variation in the AST. The first-order difference was determined from the original data to remove the autocorrelation, and it met the requirement of sample independence. The factors without collinearity were used for the driving force analysis using linear regression (linear model) and random forest regression (nonlinear model). We used the selected driving factors to establish the linear regression, the random forest model for predicting the AST, and cross-validation for verifying the prediction accuracy. Furthermore, the prediction outcomes were compared with the simplest ARIMA time-series model (control model). Our findings showed that the changing trend and the mutation of the AST were different in the UMRYR during the past 20 years. However, after the first-order difference of the AST, the amount of interannual variation in the annual sediment transport (ΔAST) was almost unchanged in the UMRYR. The five driving factors were chosen to establish the prediction models of linear regression and random forest regression, respectively. Compared with the control model, ARIMA, the prediction accuracy of the random forest model was the highest.

**Keywords:** sediment transport; driving factors; trend prediction; Yellow River; Google Earth Engine



## 1. Introduction

In recent years, owing to the influence of climate warming and human activities, the water cycle process in the Yellow River Basin (YRB) has changed significantly. As the second largest river in China, the Yellow River has long been known for having low water and abundant sediment [1]. With the large-scale implementation of water conservation projects, along with soil and water conservation projects in the past 20 years, the runoff and sediment transport have changed significantly in the upper and middle reaches of the Yellow River (UMRYR) [2,3]. These changes have led to prominent problems, such as the unbalanced spatiotemporal distribution of the water resources in the YRB [4,5]. Studying the spatiotemporal variation characteristics of the factors driving the annual sediment transport (AST) in the UMRYR is the basis for comprehensive management of the YRB, together with providing technical support for the implementation of soil and water conservation projects in the UMRYR.

Many studies have shown that the annual runoff and sediment transport in the UMRYR have experienced a significant downward trend in the last few years [4,5]. In their



analyses of the driving factors responsible for the downward trend in sediment transport in the UMRYR, scholars have expressed different views. According to research by Hu et al. [6], the main causes for the decrease in sediment transport in the YRB from 1986 to 2005 were human activities and natural environmental changes. Yao et al. [7,8], Xu et al. [9], Shi et al. [5], and Gu et al. [10] demonstrated that precipitation had a significant impact on the runoff and the sediment transport when they studied how the climate and human activities affected these phenomena in the UMRYR. However, Shretha et al. [11] demonstrated that there is a connection between increased air temperatures and sediment accumulation. Additionally, according to Yin et al. [12], vegetation cover and land type accounted for 35% of the soil erosion, which in turn affected the sediment transport in the Yellow River. In the study by Luo et al. [13], an analysis of the spatiotemporal changes among the sediment transport and the driving factors was presented. They found that precipitation, the normalized difference vegetation index (NDVI), and net primary production (NPP) each had a significantly positive correlation with sediment transport in the UMRYR.

Many scholars have shown that changes in land type have a great influence on the spatial distribution of sediment transport along the Yellow River. According to Wang et al. [14], the AST at the Tongguan gauge station will be 283, 313, and 412 Mt yr$^{-1}$ during the next 10, 20, and 50 years, respectively. According to a large number of existing research results, it could be argued that the process of sediment transport in the YRB had highly nonlinear characteristics. However, commonly used methods only offer a qualitative understanding of the changes in sediment transport, and do not yet provide quantitative predictions [15]. Yu et al. [16] used the Soil and Water Assessment Tool (SWAT) and the Coupled Model Intercomparison Project Phase 5 (CMIP5) to predict the changes in the runoff and sediment transport at the Tangnaihai gauge station. In addition, Gao et al. [17] used gray system theory to predict the long-term trend in the runoff and sediment transport in the UMRYR. They discovered that the predicted values for the runoff and sediment transport were greater than the actual values, indicating that the long-term forecasting of runoff and sediment transport was still at the development and exploration stage. In these studies, the effect of changes in the climatic factors and human activities on the sediment transport in the Yellow River was confirmed, but the driving factors considered were relatively simple. At the same time, since the prediction models of the sediment transport in the UMRYR were too complex and their prediction results were not sufficiently accurate, it was not possible to apply these models widely.

In order to resolve this problem, we used the Google Earth Engine (GEE) platform to extract historical remote sensing data on the potential driving factors, such as vegetation, soil moisture, precipitation, land type, and population, among others. The correlations between the driving factors and their correlation with the amount of interannual variation in annual sediment transport (ΔAST) were analyzed simultaneously, using the Spearman correlation combined with stepwise regression for collinearity diagnosis. The driving factors without collinearity were applied in the driving force analysis using linear regression (linear model) and random forest regression (nonlinear model). Based on the results of the analysis of the driving force, a linear and nonlinear regression model was established to predict the AST in the UMRYR. Our research aimed to establish a simple, feasible, and accurate model to predict the changes in the AST in the UMRYR and reveal the factors affecting the sediment variation along the Yellow River.

## 2. Study Area

The Yellow River, with a total length of 5464 km, and a total basin area of approximately 800,000 km$^2$, originates in the Bayan Har Mountains, and eventually, flows into the Bohai Sea. The UMRYR account for 91% of the total area of the YRB. Our study area (94–114 E, 32–42 N) covers five main gauge stations along the Yellow River: Tangnahai, Lanzhou, Toudaoguai, Longmen, and Tongguan. The elevation is between 258 and 6254 m (Figure 1). The characteristics of the UMRYR include unconsolidated soil, broken terrain, a low vegetation coverage rate, and an uneven distribution of precipitation [15,18]. The

upper reaches have an area of 386,000 km², accounting for 51.3% of the YRB, located in arid and semiarid areas, with average temperatures ranging from 1 to 6 °C, and average annual precipitation of 105–756 mm. The middle reaches are located in semi-humid and semiarid monsoon climate regions, with average annual precipitation of 320–836 mm, and average temperatures between 7 and 11 °C. The area of the middle reaches accounts for 45.7% of the total area of the YRB, but the sediment transport in this area accounts for 92% of the YRB total. The area is characterized by large volumes of coarse sand, and it is one of the most frequent sites of rainstorms in China. In the UMRYR, the spatial distribution of precipitation is extremely uneven, decreasing from southeast to northwest.

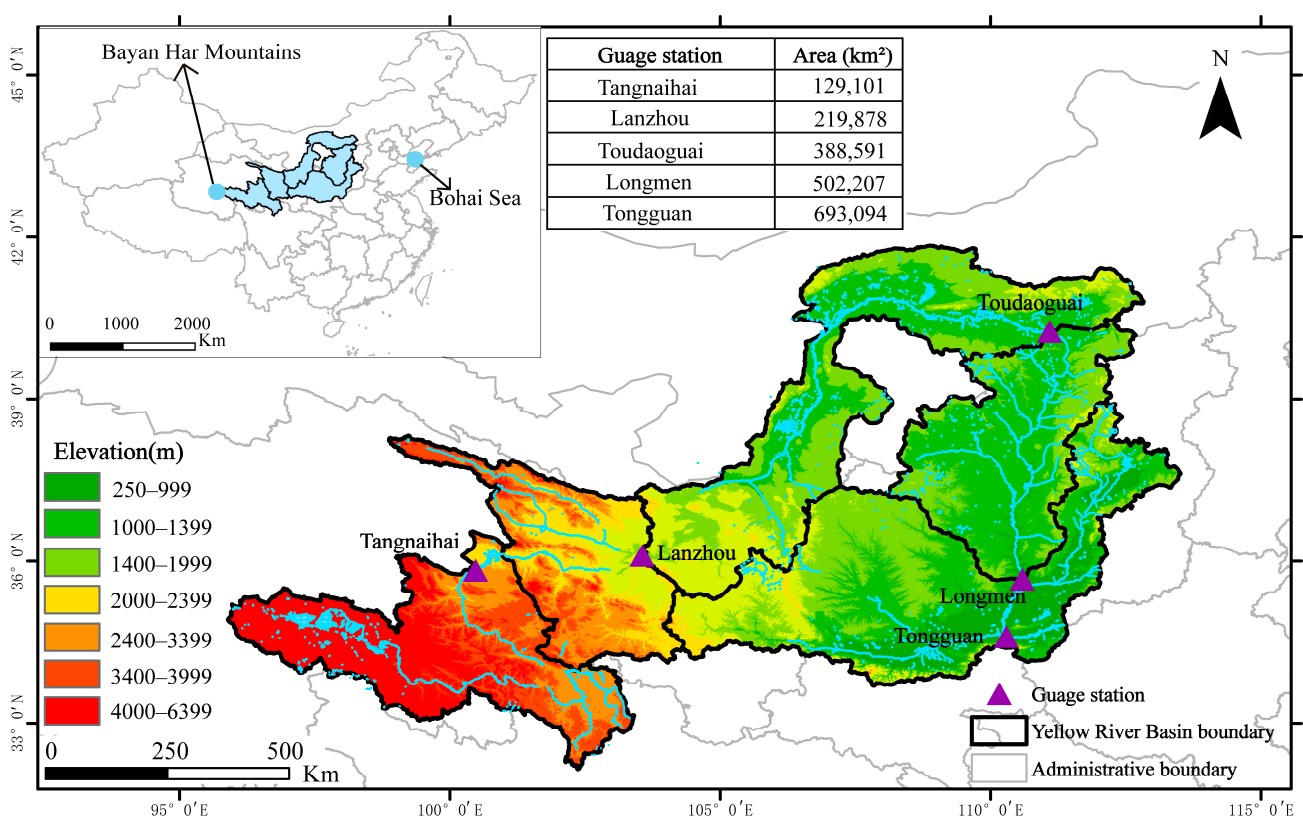

| Guage station | Area (km²) |
| --- | --- |
| Tangnaihai | 129,101 |
| Lanzhou | 219,878 |
| Toudaoguai | 388,591 |
| Longmen | 502,207 |
| Tongguan | 693,094 |

**Figure 1.** Topographic map and gauge station distribution in the study area.

## 3. Materials and Methods

### 3.1. Data

Using judgment based on professional knowledge, the 15 driving factors that might affect the AST of the UMRYR were selected. These 15 driving factors were derived from the Google Earth Engine (GEE) platform (https://earthengine.google.com, accessed on 5 November 2022), and all the driving factors data pre-processing was conducted on the GEE platform. For example, the remote sensing image was clipped according to the reaches, and the annual precipitation was averaged pixel by pixel, etc. We obtained 100 samples through 20 years of observations in five reaches. The MOD13Q1 V6 satellite data with a spatial resolution of 250 m and a temporal resolution of 16 days were used to obtain the enhanced vegetation index (EVI) and NDVI data. The normalized difference water index (NDWI) data were derived from the Landsat 5 and Landsat 8 datasets. The NPP data were from the Landsat net primary production CONUS dataset, with a spatial resolution of 30 m. The summer precipitation (SP) data and the soil moisture (SM) data were derived primarily from the CHIRPS daily dataset and the ERA5-Land dataset, and their spatial resolutions were 5566 m and 11,132 m, respectively. The population (POP) data were derived from the WorldPop global project population dataset, which has a spatial resolution of 100 m and a temporal resolution of 1 year. The land type (water body (WB), forest (FO), shrubland (SL),

and farmland (FL)) data came from the MCD12Q1.006 dataset, with a spatial resolution of 500 m. The DMSP–OLS dataset was used for the night light (OLS) data, and its spatial resolution was 927.67 m.

The AST data for the five gauge stations were obtained from the River Sediment Bulletin of China (http://xxzx.mwr.gov.cn, accessed on 20 November 2022). The River Sediment Bulletin of China is an official endorsement. The daily runoff and sediment data were measured by the basin organization using sediment runoff meters within the control range of the reaches, and were published in the River Sediment Bulletin of China every year, which has an official endorsement. For example, Hu et al. [2], Liu et al. [5], and Yao et al. [7] used water and sediment data from the River Sediment Bulletin of China to study the temporal and spatial variation trends of the water and sediment in the YRB. They also analyzed the driving forces behind these trends. The vector boundaries used in this study were obtained from the National Earth System Science Data Center, National Science and Technology Infrastructure of China (https://www.geodata.cn, accessed on 20 November 2022). Table 1 shows the sources and abbreviations for the data.

**Table 1.** Driving factors and data sources.

| | Value (Abbreviation) | Unit | Data Source |
|---|---|---|---|
| 1 | Annual sediment transport (AST) | Mt yr$^{-1}$ | The River Sediment Bulletin of China (2001–2020) |
| 2 | Normalized difference vegetation index (NDVI) | / | MOD13Q1 V6 |
| 3 | Enhanced vegetation index (EVI) | / | MOD13Q1 V6 |
| 4 | Normalized difference water index (NDWI) | / | Landsat 5 and 8 |
| 5 | Net primary production (NPP) | Kg C/m$^2$ | Landsat Net Primary Production CONUS |
| 6 | Population (POP) | individuals | WorldPop Global Project Population Data |
| 7 | Soil moisture 0–7 cm (SM(0–7)) | m$^3$/m$^3$ | ERA5-Land Monthly Averaged—ECMWF Climate Reanalysis |
| 8 | Soil moisture 7–28 cm (SM(7–28)) | m$^3$/m$^3$ | ERA5-Land Monthly Averaged—ECMWF Climate Reanalysis |
| 9 | Soil moisture 28–100 cm (SM(28–100)) | m$^3$/m$^3$ | ERA5-Land Monthly Averaged—ECMWF Climate Reanalysis |
| 10 | Soil moisture 100–289 cm (SM(100–289)) | m$^3$/m$^3$ | ERA5-Land Monthly Averaged—ECMWF Climate Reanalysis |
| 11 | Water body (WB) | % | MCD12Q1.006 |
| 12 | Forest (FO) | % | MCD12Q1.006 |
| 13 | Summer precipitation (SP) | mm | CHIRPS Daily |
| 14 | Night light (OLS) | nanoWatts/cm$^2$/sr | DMSP OLS |
| 15 | Shrubland (SL) | % | MCD12Q1.006 |
| 16 | Farmland (FL) | % | MCD12Q1.006 |

*3.2. Research Methods*

3.2.1. Mann–Kendall Mutational Test

Linear fitting was performed on the AST data from the five gauge stations to determine the variation trend in AST for the past 20 years. The Mann–Kendall (MK) test [19] is a method commonly used in meteorology to study mutations. In this study, the MK mutation test was used to analyze the mutation of the AST in the UMRYR. We supposed that time series $(x_1, x_2, \ldots, x_n)$ existed. The specific method is as follows. $S_k$ is defined as Equation (1):

$$S_k = \sum_{i=1}^{k} \sum_{j=1}^{i-1} \alpha_{ij} \ (k = 2, 3, 4, \ldots, n) \tag{1}$$

$$\alpha_{ij} = \begin{cases} 1 & x_i > x_j \\ 0 & x_i \leq x_j \end{cases} 1 \leq j \leq i \tag{2}$$

The statistic $UF_k$ is defined as Equation (3):

$$UF_k = S_k - E(S_k)/\sqrt{var(S_k)} \ (k = 1, 2, 3, \ldots, n) \tag{3}$$

$$E(S_k) = k(k-1)/4 \tag{4}$$

$$var(S_k) = k(k-1)(2k+5)/72 \tag{5}$$

Time series $x$ is arranged in reverse order and calculated with Equation (3), while ensuring Equation (6):

$$\begin{cases} UB_k = -UF_{k\prime} \\ k' = n+1-k \end{cases} (k = 1,\ 2,\ \ldots,\ n) \tag{6}$$

By analyzing statistical series $UF_k$ and $UB_k$, the change trend in series $x$ can be further analyzed. The mutation time and region can be determined. If $UF_k > 0$, the data sequence is in an upward trend; if $UF_k = 0$, the data sequence has no changing trend; if $UF_k < 0$, the data sequence is in a downward trend. When the $UF_k$ exceed the significant critical values, they show an obvious increasing or decreasing trend. If an intersection point is present between the curves of $UF_k$ and $UB_k$, and falls between the credibility lines, the corresponding time of the intersection point is the starting moment of the mutation [20].

3.2.2. Driving Force Analysis Method

(1)   Spearman correlation analysis

Since the AST is a time series, there is a certain autocorrelation. The first–order difference method was used in this study to reduce the influence of autocorrelation and transform the AST data into ΔAST data. The Spearman correlation coefficient was used to study the correlations among 16 variables, because the distribution of the variables was unknown [21]. The Spearman correlation coefficient $r_s$ is computed as follows:

$$r_s = cov(rgx_i, rgy_i)/\sigma_{rgx_i}, \sigma_{rgy_i} \tag{7}$$

For a sample of size $n$, the original data $x_i$ and $y_i$ are converted into ranks $rgx_i$ and $rgy_i$, respectively, which are the ascending orders of $x_i$ and $y_i$. The $cov(rgx_i, rgy_i)$ is the covariance of the rank variables $\sigma_{rgx_i}$, and $\sigma_{rgy_i}$ are the standard deviations of the rank variables. However, a multiple regression model that involves more than one driving factor often contains the problem of multicollinearity. There are several ways to determine whether there is a multicollinearity:

(a)   The value of the correlation (the Spearman correlation between the driving factors). A high correlation value between the two driving factors shows that there is a linear relationship, and indicates that there may be a problem of collinearity [22].

(b)   The value of the variance inflation factor (VIF) is used as a criterion to detect the presence of multicollinearity in a multiple linear regression. When the value of the VIF is greater than 3, there may be a problem of multicollinearity [23].

(2)   Linear model

In this study, a stepwise regression model (SRM) was used to analyze the linear driving force of the ΔAST. The driving variables in the SRM were selected, one by one, from a group of potential driving factors without multicollinearity, based on the Akaike information criterion (AIC). After screening the driving factors, the prediction model for ΔAST was established by using multiple linear regression (MLR). All analyses were conducted using RStudio software (https://cran.r-project.org, accessed on 4 November 2022).

(3)   Nonlinear model

A random forest model (RFM) was used to analyze the nonlinear driving force of the ΔAST, and the driving factors without multicollinearity were ranked according to the most important value for each driving factor. The number of decision trees, the maximum depth of trees, and the maximum number of leaf nodes in the model were set to 100, 10, and 50, respectively, in the R program. After training, the driving factors were selected based on the importance of the IncNodePurity value. After the nonlinear driving factors were screened out, the driving factors were used for modeling and prediction.

### 3.2.3. Accuracy Validation and Prediction of Models

(1)    Model accuracy validation

Cross-validation was used to evaluate and select the model in this study [23]. The two prediction models (MLR and RFM) were 5–fold cross–validated. In this method, the dataset was shuffled randomly, after which it was partitioned into *k* groups. In this case, we considered the value of *k* as 5. Of these folds, one was considered the testing dataset, and the others were considered the training dataset. The boosted classifier was then fitted to the training dataset and the evaluation was processed on the testing dataset. For each of these folds, the evaluation scores were accumulated, and the mean score was calculated as the final evaluation score.

(2)    Prediction of AST

The model with the highest accuracy was obtained by the cross–validation (comparison between the MLR and RFM), and the ΔAST at the five gauge stations during the following 3 years was predicted. Next, through the R language, the ΔAST was restored to the AST by using the first–order difference restoration code. The results were compared with the simplest ARIMA, which only considered the AST time series, and did not consider the driving factors.

ARIMA is a popular and simple method of analysis used in the prediction of time series. In the ARIMA model, the future value of a variable is a linear combination of past values and past errors, expressed as follows:

$$Y_t = \varnothing_0 + \varnothing_1 Y_{t-1} + \varnothing_2 Y_{t-2} + \ldots + \varnothing_p Y_{t-p} + \varepsilon_t - \theta_1 \varepsilon_{t-1} - \theta_2 \varepsilon_{t-2} - \ldots - \theta_q \varepsilon_{t-q} \quad (8)$$

where $Y_t$ is an actual value and $\varepsilon_t$ is a random error at *t*; $\varnothing_p$ and $\theta_q$ are coefficients; and *p* and *q* are integers that are often referred to as the autoregressive and moving average, respectively [24]. The technical flowchart for this paper is shown in Figure 2.

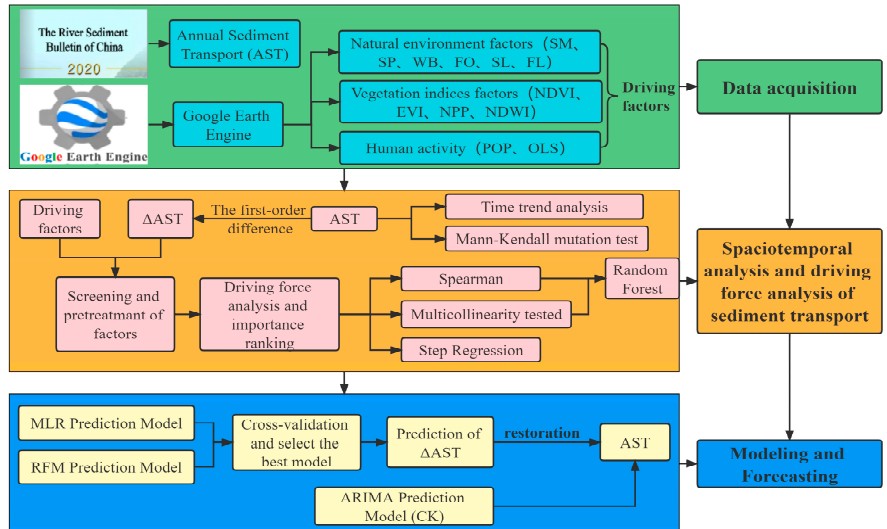

**Figure 2.** Flowchart showing the analysis of the factors driving ΔAST in the UMRYR and the trend prediction of AST. The abbreviations, AST and ΔAST, denote the annual sediment transport and amount of the interannual variation in the annual sediment transport, respectively. Table 1 shows the short meanings of the variables. The abbreviations, MLR and RFM, denote the multiple linear regression and random forest model, respectively.

## 4. Results

### 4.1. Analysis of Variation Trend and Mutation of the AST

Through the linear fitting (Figure 3), we found that during the past 20 years, the variation trend in the AST in the UMRYR has not remained stable. The AST in the upper reaches

of the Yellow River (URYR), from the Tangnaihai to the Toudaoguai gauge station, showed an upward trend (Figure 3a–c). The annual change rates of the AST were 0.355 Mt yr$^{-1}$ ($p > 0.05$), 0.371 Mt yr$^{-1}$ ($p < 0.05$), and 3.423 Mt yr$^{-1}$ ($p < 0.05$) at the Tangnaihai, Lanzhou, and Toudaoguai gauge stations, respectively. However, the AST in the middle reaches of the Yellow River (MRYR), from the Toudaoguai to the Tongguan gauge station, showed a downward trend (Figure 3d,e). The annual change rates of the AST were 3.565 Mt yr$^{-1}$ ($p > 0.05$), and 12.771 Mt yr$^{-1}$ ($p < 0.05$) at the Longmen and Tongguan gauge stations, respectively.

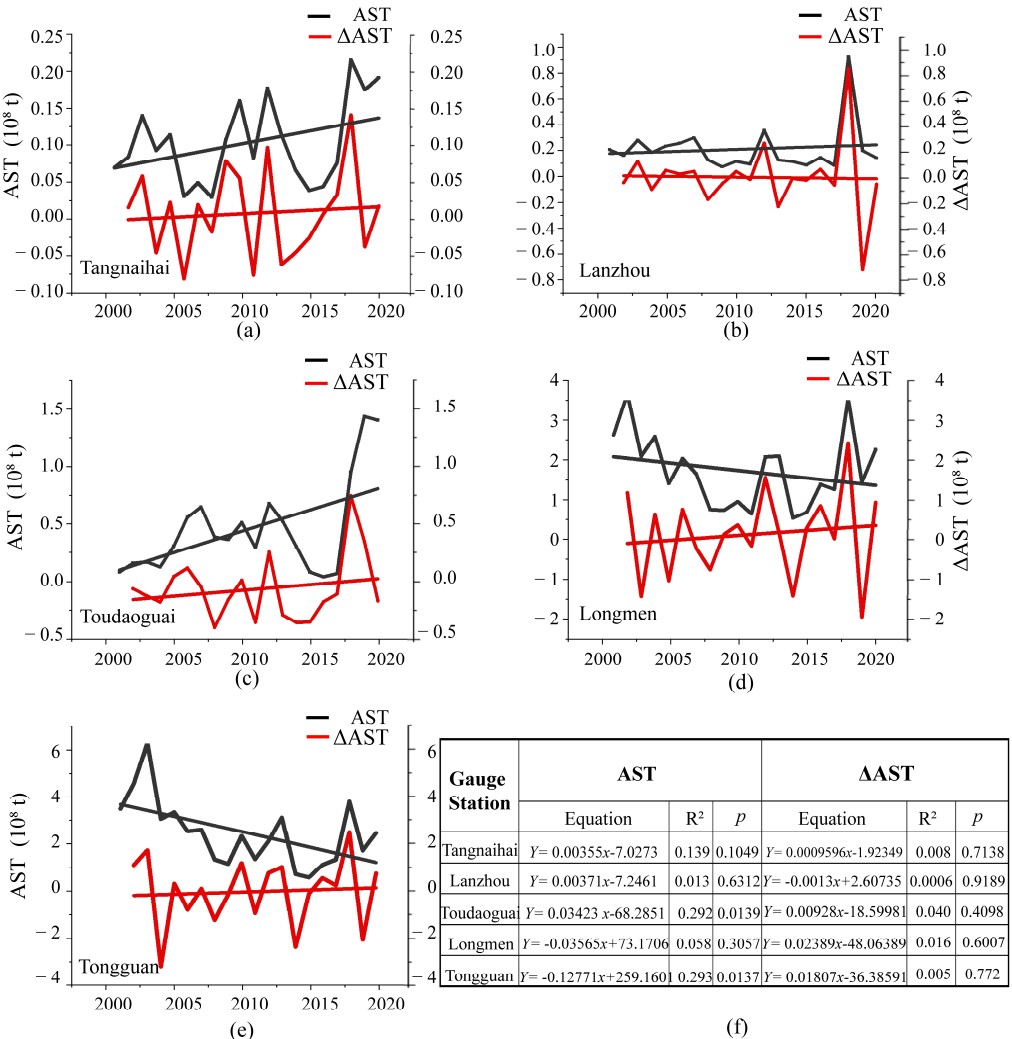

| Gauge Station | AST | | | ΔAST | | |
|---|---|---|---|---|---|---|
| | Equation | R² | p | Equation | R² | p |
| Tangnaihai | $Y= 0.00355x-7.0273$ | 0.139 | 0.1049 | $Y= 0.0009596x-1.92349$ | 0.008 | 0.7138 |
| Lanzhou | $Y= 0.00371x-7.2461$ | 0.013 | 0.6312 | $Y= -0.0013x+2.60735$ | 0.0006 | 0.9189 |
| Toudaoguai | $Y= 0.03423x-68.2851$ | 0.292 | 0.0139 | $Y= 0.00928x-18.59981$ | 0.040 | 0.4098 |
| Longmen | $Y= -0.03565x+73.1706$ | 0.058 | 0.3057 | $Y= 0.02389x-48.06389$ | 0.016 | 0.6007 |
| Tongguan | $Y= -0.12771x+259.1601$ | 0.293 | 0.0137 | $Y= 0.01807x-36.38591$ | 0.005 | 0.772 |

**Figure 3.** Variation trend in AST and ΔAST in the UMRYR, showing the gauge stations in the URYR (**a**–**c**) and the gauge stations in the MRYR (**d**,**e**), and the equation for the variation trend in the AST and ΔAST in the UMRYR (**f**). The abbreviations, AST and ΔAST, denote the annual sediment transport and the amount of interannual variation in the annual sediment transport, respectively. The abbreviations, UMRYR and MRYR, denote the upper and the middle reaches of the Yellow River, respectively.

At the same time, after the first–order difference of the AST, the ΔAST was almost unchanged in the UMRYR (red line in Figure 3), and the $p$ values were all greater than 0.05. The temporal trend influence of the AST in the UMRYR was effectively eliminated by the first–order difference. The driving factors in this study were treated in the same way. The processed data (ΔAST) had no autocorrelation, therefore, they were used for the statistical analysis.

Figure 4a shows the Friedman test and violin plots of the AST and ΔAST data at each gauge station. It can be observed that the AST in the UMRYR differed significantly

($\chi^2 = 73.84$, $p < 0.001$). Multiple comparisons showed that there was a significant difference between the Tangnaihai and Toudaoguai gauge stations in the URYR ($p = 0.002$). Similarly, there was a significant difference between the Toudaoguai and Tongguan gauge stations in the MRYR ($p = 0.001$). However, after the first difference, there was no difference in the $\Delta$AST at each gauge station (Figure 4b, $\chi^2 = 1.425$, $p < 0.840$). The $\Delta$AST data for the five gauge stations met the requirement of sample independence. The linear trend in the AST data was removed by the first-order difference.

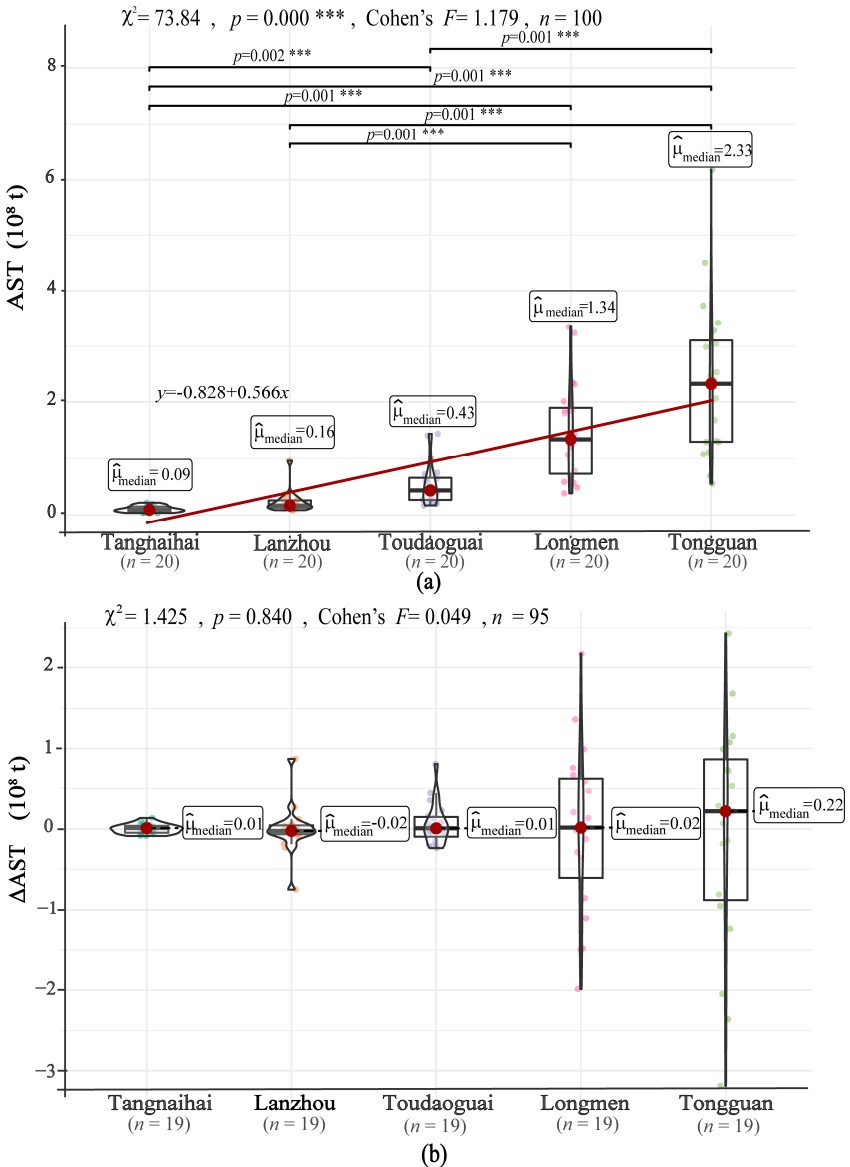

**Figure 4.** Friedman test of AST (**a**) and $\Delta$AST (**b**) at the five gauge stations. The abbreviations, AST and $\Delta$AST, denote the annual sediment transport and the amount of interannual variation in the annual sediment transport, respectively. The red line in (**a**) is the trend line for the annual mean sediment transport (2001–2020) at each gauge station. The symbol '***' indicates that there is a significant difference between the two gauge stations.

According to the Mann–Kendall mutation analysis of the AST for the past 20 years, the mutations occurred more frequently in the URYR (Figure 5a–c) than in the MRYR (Figure 5d,e). In the past twenty years, there were two or three mutation points within the gauge stations in the URYR. However, regarding the gauge stations within the MRYR, there was only one mutation point at the Longmen gauge station, and no mutation point at

the Tongguan gauge station (the intersection point was not within the confidence interval). Through the change in the $UF_k$ value, we can also understand in detail the changing trend in the AST. In Figure 5a–c, the value of $UF_k$ fluctuated significantly around zero. Although the overall trend in the AST increased (Figure 3a–c), there was also a downward trend in some years, and the trend fluctuated greatly in the URYR. By contrast, the value of the $UF_k$ was mostly below zero (Figure 5d,e) in the MRYR, indicating that the overall trend in the AST decreased (Figure 3d,e), but there was also an upward trend during the first few years. In addition, it can also be seen in Figure 5f that the cumulative amount of AST at the gauge stations in the MRYR (Longmen and Tongguan) was significantly higher than that of the gauge stations in the URYR. This indicates that the AST along the Yellow River was mainly concentrated in the MRYR.

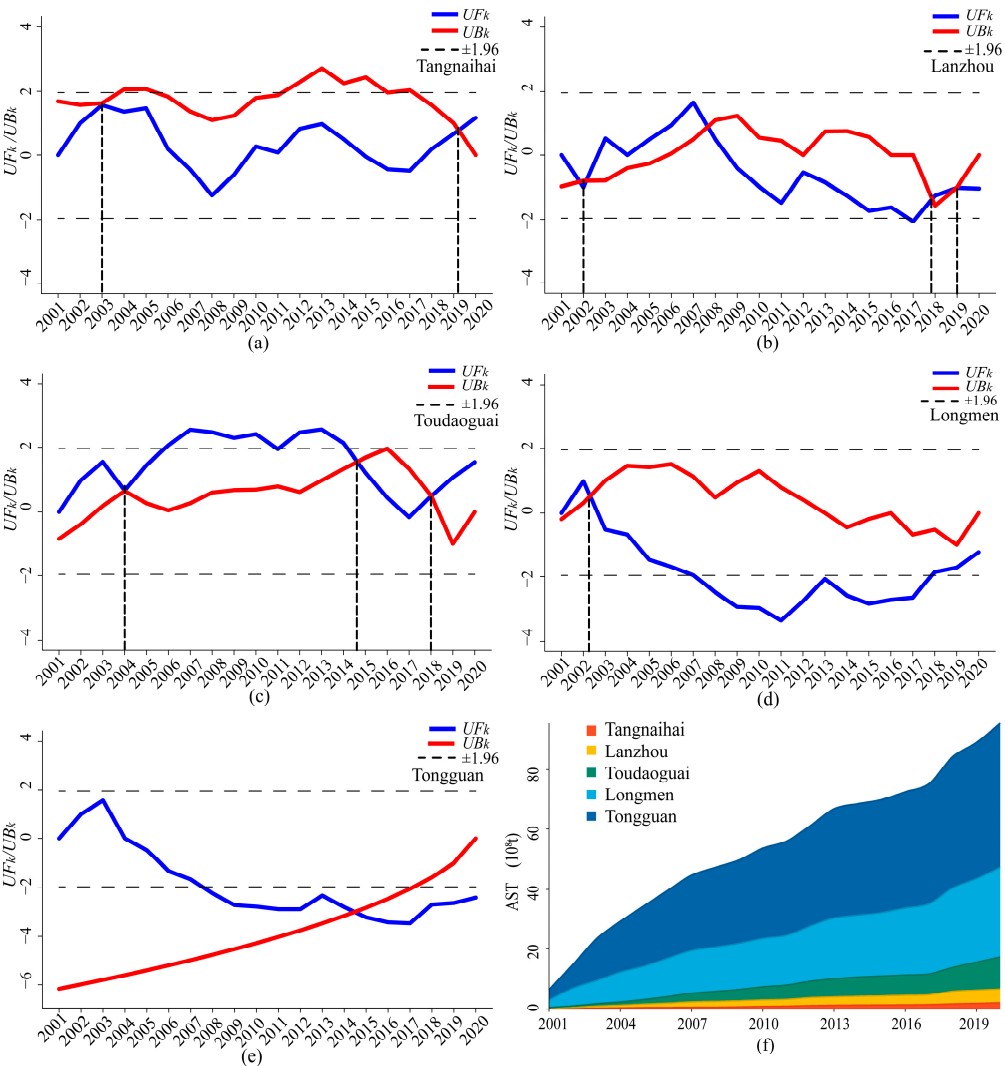

**Figure 5.** Mann–Kendall mutation test of the AST: URYR (**a**–**c**), MRYR (**d**,**e**), and a graph showing the cumulative amount of AST for the five gauge stations (**f**). The abbreviation AST denotes the annual sediment transport. The abbreviations, URYR and MRYR, denote the upper and the middle reaches of the Yellow River, respectively.

### *4.2. Driving Force Analysis*

#### 4.2.1. Spearman Correlation Analysis

In this study, there were 16 variables, including dependent variables (ΔAST). The first difference was applied to all the data (in Figure 6, the prefix 'Δ' indicates the amount of interannual variation in the data, that is, the first-order difference). As shown in Figure 5,

there was a strong positive correlation between the factors reflecting the vegetation status (i.e., ΔNDVI, ΔEVI, and ΔNPP) and the ΔAST, with correlation coefficients of 0.591, 0.538, and 0.429, respectively. This was followed by the factors reflecting the environment and climate (i.e., ΔSM (28–100) and ΔSP) with correlation coefficients of 0.346 and 0.282, respectively. Additionally, it was found that the correlation coefficients between the ΔNDVI and ΔEVI, ΔSM (0–7) and ΔSM (7–28) were 0.931, and 0.941, respectively. It can be argued that there were collinearity problems between them.

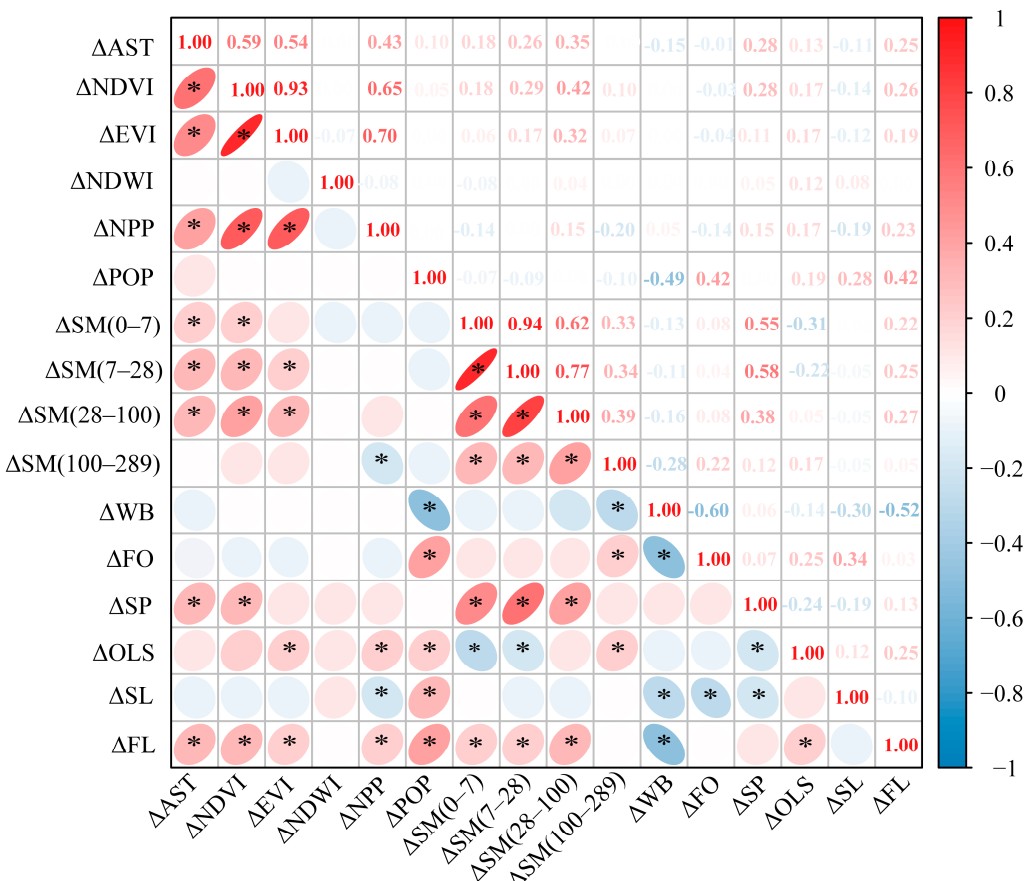

**Figure 6.** Spearman correlation analysis of ΔAST in the UMRYR. The prefix '∆' indicates the amount of interannual variation in the data. In particular, the symbol '*' indicates that there is a significant relationship between the two variables ($p < 0.1$). Refer to Table 1 for the abbreviated meanings of the variables.

### 4.2.2. Multicollinearity Test

The amount of the interannual variation in the data needed to be tested for multicollinearity before creating the RFM (nonlinear model) driving force analysis, as shown in Table 2. The VIF values of the ΔNDVI, ΔEVI, ΔNPP; ΔSM (0–7), ΔSM (7–28), ΔWB, and ΔFO; and ΔFL were greater than 3, implying potential collinearity between these driving factors. As a result, eight driving factors were finally added into the RFM (nonlinear model): ΔNDVI, ΔNDWI, ΔNPP, ΔSM (100–289), ΔSP, ΔOLS, ΔSL, and ΔFL, when combined with the results of the Spearman correlation analysis.

**Table 2.** Multicollinearity diagnosis of the driving factors. Dependent variable: ΔAST. The definitions of the abbreviated variables can be found in Table 1.

| Model | Unstandardized Coefficients | | Standardized Coefficients | *t* | *p* | Collinearity Statistics | |
|---|---|---|---|---|---|---|---|
| | *B* | *SE* | *Beta* | | | Tolerance | VIF |
| (Intercept) | −0.058 | 0.107 | | −0.538 | 0.592 | | |
| ΔNDVI | 7.426 | 13.316 | 0.220 | 0.558 | 0.579 | 0.044 | 22.857 |
| ΔEVI | 5.246 | 13.101 | 0.151 | 0.400 | 0.690 | 0.048 | 20.733 |
| ΔNDWI | 0.533 | 1.137 | 0.043 | 0.469 | 0.641 | 0.827 | 1.210 |
| ΔNPP | −2.367 | 6.609 | −0.054 | −0.358 | 0.721 | 0.299 | 3.350 |
| ΔPOP | 0.000 | 0.000 | −0.158 | −1.359 | 0.178 | 0.503 | 1.987 |
| ΔSM (0–7) | 4.199 | 26.598 | 0.046 | 0.158 | 0.875 | 0.080 | 12.430 |
| ΔSM (7–28) | 9.073 | 31.783 | 0.080 | 0.285 | 0.776 | 0.086 | 11.564 |
| ΔSM (28–100) | 1.466 | 3.648 | 0.035 | 0.402 | 0.689 | 0.876 | 1.142 |
| ΔSM (100–289) | −55.202 | 16.815 | −0.324 | −3.283 | 0.002 | 0.700 | 1.429 |
| ΔWB | −96.096 | 128.584 | −0.167 | −0.747 | 0.457 | 0.137 | 7.281 |
| ΔFO | 70.096 | 192.925 | 0.059 | 0.363 | 0.717 | 0.262 | 3.814 |
| ΔFL | 50.005 | 150.836 | 0.080 | 0.332 | 0.741 | 0.118 | 8.475 |
| ΔSP | 0.004 | 0.003 | 0.214 | 1.578 | 0.119 | 0.370 | 2.703 |
| ΔOLS | 0.528 | 0.168 | 0.304 | 3.141 | 0.002 | 0.727 | 1.375 |
| ΔSL | −16.074 | 232.564 | −0.008 | −0.069 | 0.945 | 0.553 | 1.810 |

### 4.2.3. Stepwise Regression

An SRM (linear model) analysis was conducted using the RStudio software, and the model with the lowest AIC value was selected based on the 15 previously identified ΔAST driving factors that may have affected the ΔAST in the UMRYR. Table 3 presents the results of the analysis to determine the primary driving factors that affected the changes in the ΔAST. The results of the SRM analysis indicated that ΔNDVI, ΔSM (100–289), ΔWB, ΔSP, and ΔOLS ($p < 0.05$) were the significant driving factors affecting the ΔAST in the UMRYR. Furthermore, the VIF values of the five driving factors were less than 3, indicating the absence of multicollinearity between them. Additionally, the *p* value for the equation was less than 0.05, indicating that the equation was significant and had a fit of $R^2 = 0.445$ (Table 3).

**Table 3.** Stepwise regression analysis. Dependent variable: ΔAST. The definitions of the abbreviated variables can be found in Table 1.

| Model | Unstandardized Coefficients | | Standardized Coefficients | *t* | *p* | Collinearity Statistics | |
|---|---|---|---|---|---|---|---|
| | *B* | *SE* | *Beta* | | | Tolerance | VIF |
| (Intercept) | −0.161 | 0.076 | | −2.127 | 0.036 | | |
| ΔNDVI | 12.388 | 3.000 | 0.368 | 4.129 | 0.000 | 0.812 | 1.232 |
| ΔOLS | 0.481 | 0.144 | 0.277 | 3.332 | 0.001 | 0.929 | 1.076 |
| ΔSP | 0.005 | 0.002 | 0.247 | 2.854 | 0.005 | 0.858 | 1.166 |
| ΔSM (100–289 cm) | −44.362 | 14.160 | −0.261 | −3.133 | 0.002 | 0.930 | 1.076 |
| ΔWB | −100.358 | 47.795 | −0.174 | −2.100 | 0.039 | 0.936 | 1.068 |
| | Multiple R-squared: 0.445 | | | | Adjusted R-squared: 0.407 | | |
| | F-statistic: 11.75 | | | | *p*-value: $1.236 \times 10^{-9}$ | | |

### 4.2.4. RFM Regression

Following the results of the multicollinearity analysis, an RFM (nonlinear model) analysis was conducted to analyze the driving forces and rank their importance (Figure 7). The results showed that the ΔNDVI was the most important driving factor in predicting the ΔAST in the UMRYR, followed by the ΔOLS and ΔNPP, which were very significant ($p < 0.01$) influences on the ΔAST, and ΔSP and ΔSM (100–289) were significant ($p < 0.05$).

The effects of the ΔNDWI, ΔFL, and ΔSL on the ΔAST were not significant ($p > 0.05$), which differed from the results of the stepwise regression analysis. Additionally, the RFM model had a fit $R^2$ of 0.515, which was greater than that of the stepwise regression ($R^2 = 0.445$). Based on the driving force analysis of the ΔAST in the UMRYR, the following five variables were selected for further modeling and prediction: ΔNDVI, ΔOLS, ΔNPP, ΔSP, and ΔSM (100–289).

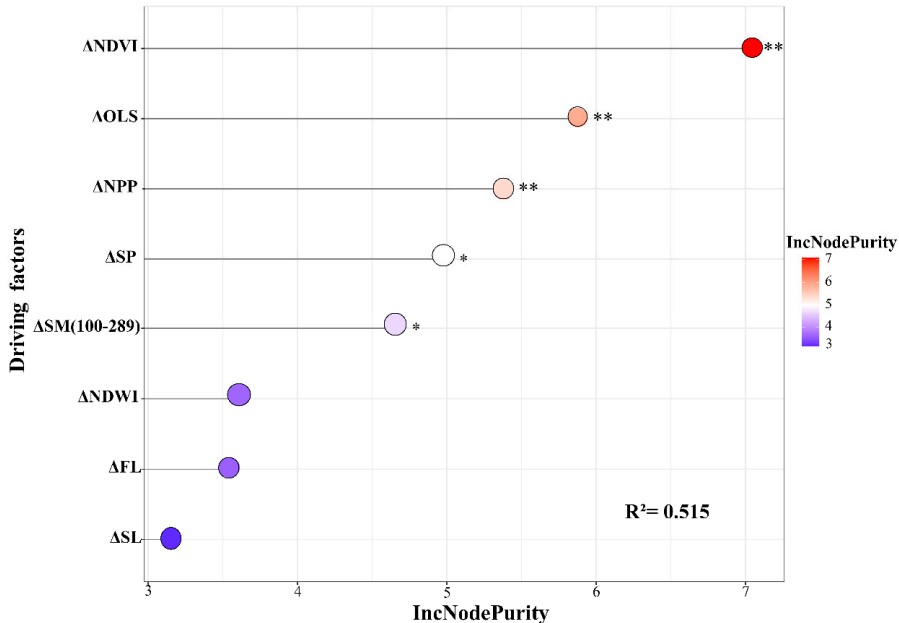

**Figure 7.** Ranking of the driving factors by importance according to the RFM results. The definitions of the abbreviated variables can be found in Table 1. The symbols '**' and '*' indicate that the factors are very significant ($p < 0.01$) and significant ($p < 0.05$), respectively. The RFM is the random forest model.

*4.3. Modeling and Prediction*

4.3.1. Cross–Validation and Selection of the Model

The driving factors obtained from the driving force analysis of the SRM and the RFM were used to establish a linear model (MLR) and a nonlinear model (RFM), respectively, to predict the ΔAST. The accuracy of the MLR and RFM models was verified by 5–fold cross–validation. To compare the prediction accuracy of the models, evaluation metrics such as $R^2$, the mean absolute error (MAE), and the root mean squared error (RMSE) were used. In Table 4, for the RFM prediction results, the evaluation metrics $R^2$, RMSE, and MAE were 0.545, 0.485, and 0.322, respectively; for the MLR, the evaluation metrics $R^2$, RMSE, and MAE, were 0.340, 1.128, and 0.875, respectively.

**Table 4.** Comparison of the model prediction accuracies. The RMSE and MAE indicate the root mean squared error and mean absolute error. The RFM and MLR indicate the random forest model and multiple linear regression model, respectively.

| Models | $R^2$ | RMSE | MAE |
|---|---|---|---|
| RFM Prediction Model | 0.545 | 0.485 | 0.332 |
| MLR Prediction Model | 0.340 | 1.128 | 0.875 |

In particular, a higher $R^2$ value indicates a better fit. Furthermore, lower values for the RMSE and MAE indicate a better fit and higher prediction accuracy. In this case, the RFM had the higher $R^2$ and the lower RMSE and MAE values, indicating that it had the better fit and higher prediction accuracy of the two models. Therefore, we used the RFM for the

final modeling and prediction of the AST, while the simplest ARIMA model was used as a control model (CK) for the RFM model.

4.3.2. Prediction of AST

Before predicting the AST, a simple forecast of the values for the five driving factors, included in the prediction model for the next three years, was performed using a linear regression method (Table 5). Before predicting the five driving factors, scatter plots were created for each driving factor, and it was found that most of them exhibited linear correlation. Therefore, the method of univariate linear regression was used for prediction. As illustrated in Figure 8, the prediction of the AST in the UMRYR uses the random forest model (RFM), established above, and the CK model (ARIMA). The forecast was divided into two parts. The left side of the black dotted line shows the forecast for the known years (from 2001 to 2020), and the right side of the black dotted line indicates the forecast for the unknown years (from 2021 to 2023) (Figure 8a–e). From the comparison of the results of the RFM prediction and the CK model prediction, it was found that the $R^2$ results for the RFM were higher for the five gauge stations than the $R^2$ results for the CK model. Furthermore, the RFM's average interpretation rate of 0.777 was higher than that for the CK model (0.318) (Figure 8f). This demonstrates that the prediction result for the RFM for the AST was better than that of the CK model. In addition, for the RFM, the goodness-of-fit order of the five gauge stations, from high to low, was: Lanzhou > Tangnaihai > Longmen > Toudaoguai > Tongguan. As a result, the nonlinear machine learning RFM was used to forecast the AST for the following 3 years. Table 6 uses the RFM to predict the specific values of the AST for the five gauge stations from 2021 to 2023. When the predictions were compared with the current measured AST for 2021, the predicted values at the Lanzhou, Longmen, and Tongguan gauge stations were close to the measured AST, and the prediction results were relatively accurate.

**Table 5.** Predicted values for the five driving factors (2021–2023). The definitions of the abbreviated variables can be found in Table 1. The prefix 'Δ' indicates the amount of interannual variation in the data.

| Gauge Stations | Year | ΔNDVI | ΔOLS | ΔNPP | ΔSP | ΔSM (100–289) |
|---|---|---|---|---|---|---|
| Tangnaihai | 2021 | −0.008469776 | 0.047935561 | 0.00049489 | −28.92988857 | −0.003606281 |
| | 2022 | 0.003678309 | 0.036382925 | 0.00022413 | −7.465995264 | 0.000653275 |
| | 2023 | 0.003789897 | 0.03863334 | −0.00004662 | −8.100876928 | 0.000702471 |
| Lanzhou | 2021 | −0.013486897 | 0.065244886 | −0.0012948 | −34.85416005 | −0.004571964 |
| | 2022 | 0.004905457 | 0.006091741 | −0.0016423 | −9.49682083 | −0.000427942 |
| | 2023 | 0.004750819 | 0.005208895 | −0.0019899 | −36.16739651 | −0.000340503 |
| Toudaoguai | 2021 | −0.02532223 | 0.079224243 | −0.0022115 | −49.84926486 | −0.002503088 |
| | 2022 | −0.0021852 | −0.107179825 | −0.0027162 | −14.74115543 | −0.001866121 |
| | 2023 | −0.00266316 | −0.114141706 | −0.0032208 | −15.27368132 | −0.001730933 |
| Longmen | 2021 | −0.03150929 | 0.074433277 | −0.0026485 | −57.20250634 | −0.002363319 |
| | 2022 | 0.024095419 | −0.149968857 | −0.0033447 | −18.36618505 | 0.00259626 |
| | 2023 | −0.02737186 | −0.159664341 | −0.0040409 | −18.97639053 | 0.004800315 |
| Tongguan | 2021 | −0.027818798 | 0.091943714 | −0.0030094 | −33.88406194 | 0.003120987 |
| | 2022 | −0.003459783 | −0.161176695 | −0.003814 | −9.50323058 | 0.000239401 |
| | 2023 | −0.00427008 | −0.175411692 | −0.0046185 | −9.723017661 | 0.000325405 |

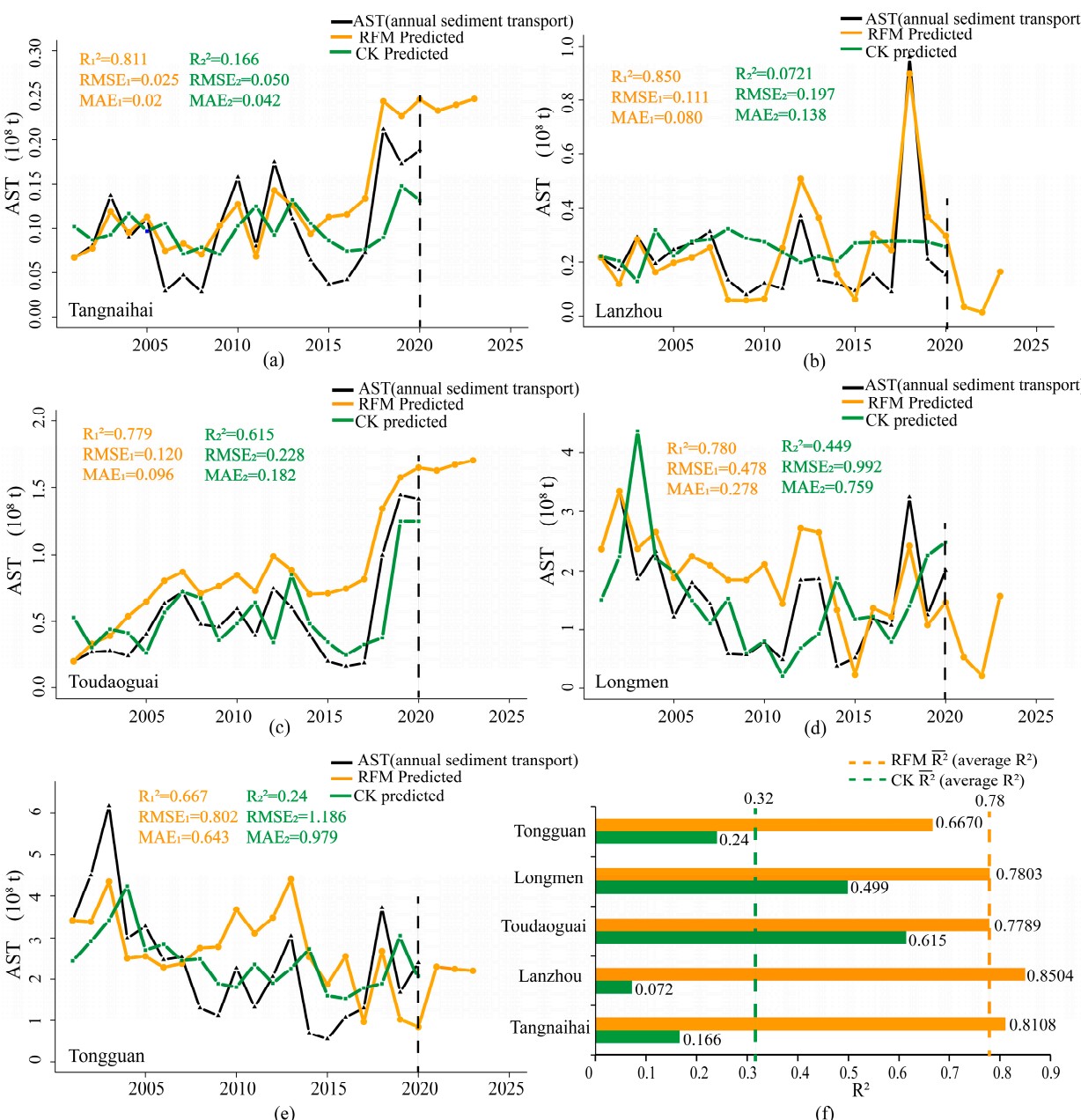

**Figure 8.** (**a–e**) The prediction of AST at the five gauge stations in the UMRYR using the RFM and ARIMA models, and a comparison with the measured AST data. The left side of the black dotted line represents the forecast for the known years (2001 to 2020), while the right side represents the forecast for the unknown years (2021 to 2023). (**f**) The prediction accuracy comparison for the five gauge stations using the RFM and ARIMA models.

**Table 6.** AST data for the five gauge stations for the next 3 years, predicted by the RFM. The abbreviation AST denotes the annual sediment transport and RFM denotes the random forest model.

| Year | Tangnaihai (Mt yr$^{-1}$) | Lanzhou (Mt yr$^{-1}$) | Toudaoguai (Mt yr$^{-1}$) | Longmen (Mt yr$^{-1}$) | Tongguan (Mt yr$^{-1}$) |
|------|------|------|------|------|------|
| 2021 | 23.249 | 3.358 | 162.733 | 52.650 | 165.658 |
| 2022 | 23.912 | 1.409 | 167.035 | 21.293 | 224.810 |
| 2023 | 24.575 | 16.460 | 170.318 | 156.870 | 220.370 |

## 5. Discussion

### 5.1. Changes in AST in the UMRYR

Through the results of this study, it was found that the AST in the UMRYR was inconsistent. The AST in the MRYR showed a significant downward trend during the past 20 years, and the AST in the URYR showed an upward trend (Figure 3). The changing trend and the mutation change in the AST in the MRYR were consistent with most research results [13,17]. However, the variation in sediment transport in the URYR is different from the results in other studies. This is because the previous studies [25,26] on AST in the URYR were conducted on a time scale of more than 40 years, and in this time series, the AST in the URYR showed a downward trend. Most scholars, such as Hu et al. [27,28], for example, have suggested that human activities have played a dominant role in the sharp reduction in the AST in the YRB, in which the main driving factors included large-scale soil and water conservation measures, such as terraced fields, check dams, and reservoir construction [29]. This was similar to the research results in this paper, because in the analysis of ranking the driving factors by importance (Figure 7), the two highest ranking driving factors (NDVI and OLS) both had a certain correlation with human activities. The Bulletin of the First National Water Resource Census (http://www.mwr.gov.cn, accessed on 13 January 2023) suggests that there are 5340 key dams in the tributaries feeding the MRYR, and that sediment fills 1090 of these dams. In addition, the reservoirs data in Figure 9 came from the Yearbook of the Yellow River (https://navi.cnki.net, accessed on 5 December 2022). It can be seen that in the past 20 years the number of reservoirs in the MRYR stabilized around a figure between 1700 and 1800, which is more than the number of reservoirs in the URYR. Furthermore, after 2000, the average AST reduction in the check dams in the MRYR was about 135 MT yr$^{-1}$, the average AST reduction in the terraces in the main sand-producing areas in the MRYR was 422 MT yr$^{-1}$, and the average AST retention volume of the reservoirs in the MRYR was 98 MT yr$^{-1}$ [30]. This proves that the implementation of soil and water conservation projects can indeed effectively reduce AST in the YRB. Therefore, in this study, it was considered that the possible reason for the inconsistent variation in the AST in the UMRYR was that there were more soil and water conservation projects in the MRYR. In addition, more mutations occurred in the AST in the URYR than in the MRYR (Figure 5). This study considered that a possible reason for this was that the mountainous terrain in the URYR is relatively fragile and prone to flash flood disasters, resulting in the occurrence of more mutations in the AST [6]. In contrast, the MRYR area is relatively gentle, with stable riverbeds and relatively stable changes in the AST.

Because the AST recorded at the five gauge stations in the UMRYR had obvious changing trends and autocorrelation, the data did not meet the statistical requirements. In time–series analysis, the first–order difference is often used as a preprocessing step to create the time–series data, eliminate the random trends they contain, and stabilize them to ensure the independence of the data samples [31]. In this research, after the first–order difference ($\Delta$AST), there was no trend, the data were, therefore, used for the statistical analysis (Figure 4).

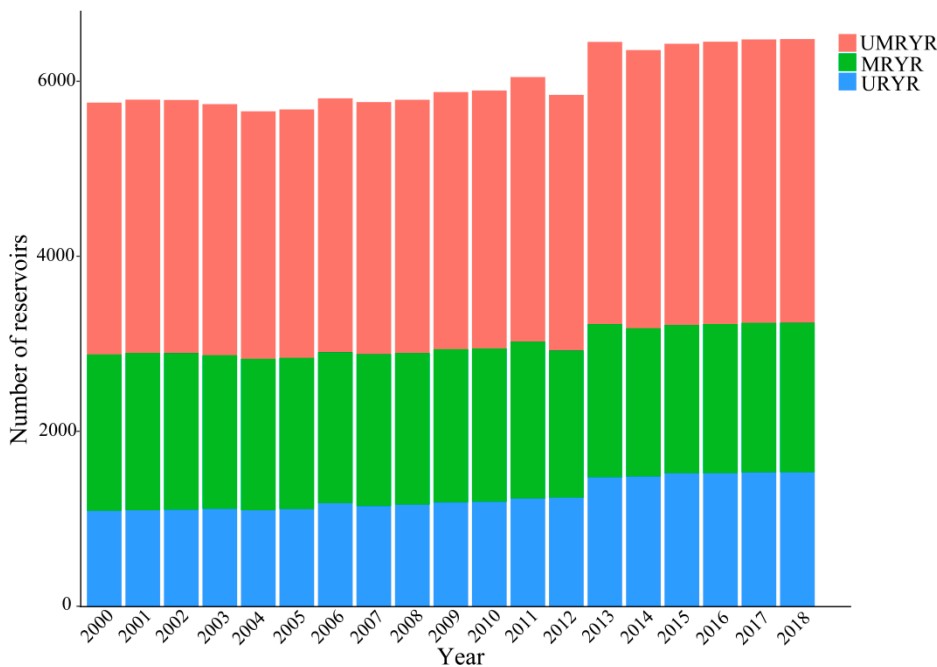

**Figure 9.** Twenty-year comparison and trend analysis of the reservoirs in the UMRYR for 2000–2018. The URYR and MRYR indicate the upper reaches of the Yellow River and the middle reaches of the Yellow River, respectively.

### 5.2. Analysis of Driving Force of ΔAST and Modeling

In this study, after analyzing the ΔAST data's linear and nonlinear driving forces, it was found that the ΔAST data were more suitable for the nonlinear driving force analysis, for which the $R^2$ (0.515) (Figure 7) was better than the $R^2$ (0.445) (Table 3) for the linear driving–force analysis. At the same time, after the 5–fold cross–validation of the RFM and MLR prediction models (Table 4), it was also found that the nonlinear driving-force analysis of the ΔAST data was a better method.

Therefore, in this study, the driving factors that had a significant effect on the ΔAST in the UMRYR were ΔNDVI, ΔOLS, ΔNPP, ΔSP, and ΔSM (100–289), obtained through the RFM. As the NDVI increases, the capability of the vegetation to protect against erosion and sediment deposition was enhanced [32]. Furthermore, other studies [13,33] showed that the annual average NDVI of the YRB increased significantly, and sediment transport decreased exponentially with increasing NDVI. According to the research results presented by Luo et al. [13], the degree of variation in the NPP was found to be most similar to that of sediment transport in the YRB, which was the same as the result presented in this paper. Both NDVI and NPP are vegetation indices. Studies have shown that changes in the water and sediment in the YRB were closely related to changes in the vegetation. When vegetation recovery reaches a certain level, the soil and water conservation function of the vegetation undergoes a qualitative change, and it plays a more significant role in conserving soil and water. In particular, when vegetation coverage exceeds 60%, soil erosion can be effectively controlled. Thus, effectively controlling the sediment transport variation in the YRB [10,14]. According to research by Shi et al. and Gu et al. [5,10], precipitation is also considered to have a significant effect on sediment transport in the UMRYR. High–intensity rainfall causes a greater change in sediment transport in the UMRYR compared to medium– and low–intensity precipitation. Moreover, the high-intensity precipitation in the UMRYR is mainly concentrated in June to September. Additionally, deep soil moisture is a highly important source of water for vegetation in arid and semiarid regions. Different types of vegetation respond differently to deep soil moisture, which is itself affected by precipitation [34]. Therefore, the SM (100–289) indirectly affects the change of the ΔAST in the UMRYR by influencing the vegetation. The OLS images were formed by the detected light

radiation induced by human activities on the Earth's surface; such images can directly reflect artificial surface regions and locations with significant human activities [35,36]. The impact of the OLS on sediment transport in the watershed is mainly due to its influence on urbanization and human activity levels, which in turn affect factors such as land use, coverage, and management, and thereby, impact hydrological cycles and sediment transport processes in the UMRYR. Additionally, the OLS can affect human behavior, such as increased traffic flow at night and nighttime entertainment activities, which can increase the likelihood of soil erosion and sediment production in the UMRYR, further impacting sediment transport rates and variations. Therefore, this study suggests that the drivers screened by the RFM to build an RFM prediction model can be used to accurately predict future AST in the UMRYR.

*5.3. Limitations and Future Work*

After analyzing previous studies [25,28], it was found that when analyzing the runoff and sediment transport in the YRB, the time range covered by most of the research was more than 40 years. In this research, the time series of the sediment transport data was short, and only the AST data were used, resulting in a small sample size, and no analysis of the periodicity of the sediment transport changes was conducted. In future research, it is recommended that data from longer time series are collected, including quarterly or monthly data, to increase the sample size and provide a more comprehensive analysis of the trend, mutation, and periodicity of sediment transport in the YRB. This will enable a more in-depth understanding of the sediment transport dynamics in the region and aid in the development of effective management strategies. According to previous research [4], human activities were responsible for 70% of the sediment transport in the UMRYR in recent years, with soil and water conservation measures contributing to 40% of this value. Large–scale soil and water conservation measures, such as terraced fields and check dams, can have a significant impact on the processes of surface water production and sediment transport in the YRB. These measures can effectively improve the ecological environment in the region, reduce soil erosion, and decrease runoff and sediment transport [37,38]. This results in a positive effect on the underlying surface characteristics, and helps to maintain a healthy and sustainable environment in the YRB.

In this study, the driving factors behind the ΔAST in the UMRYR were not sufficiently comprehensive, as they did not take into account human-induced factors such as terraced fields and check dams. Therefore, the coefficient of determination ($R^2$) for the ΔAST prediction model was relatively low, at 0.545 (Table 4). To address this limitation, future research should aim to incorporate these human-induced factors into the analysis in order to enhance the interpretability of the model.

**6. Conclusions**

This study used AST data from five gauge stations in the UMRYR and satellite remote sensing image data of the GEE to study the variation trend and mutation in the AST in the UMRYR. The ΔAST (the first–order difference of the AST) and the SRM and RFM were used to study the driving factors affecting the change in the AST in the UMRYR, and the prediction of the AST in the UMRYR was eventually completed by using the RFM.

The main conclusions in this study are that from 2001 to 2020, in the UMRYR, the AST recorded at the gauge stations in the URYR exhibited an upward trend, while the AST at the gauge stations in the MRYR displayed a downward trend, and the mutation point of the AST in the URYR was higher than that in the MRYR. This suggests that the change in the AST in the URYR was more complex than that in the MRYR. Furthermore, it was found that the cumulative amount of AST in the MRYR was significantly higher than that in the URYR. This indicates that the AST along the UMRYR was mainly concentrated in the MRYR.

Through the use of data from satellite remote sensing images and statistical analyses, it was determined that the main drivers of this change were the ΔNDVI, ΔOLS, ΔNPP,

ΔSP, and ΔSM (100–289). This study also found that the RFM established by these five driving factors was a simple, feasible, and accurate model for analyzing and predicting the changes in the AST in the UMRYR. In future studies, it is recommended that greater consideration is given to anthropogenic activities, such as the implementation of soil and water conservation measures, e.g., terraced fields and check dams, to improve the accuracy of the AST prediction. The present study provides significant perspectives on the AST dynamics in the UMRYR and demonstrates the potential of remote sensing and statistical analyses as powerful tools for identifying the underlying factors that drive changes in sediment levels.

**Author Contributions:** Conceptualization, J.W. and J.T.; methodology, J.W. and J.L.; validation, J.W., J.T. and J.L.; formal analysis, J.W. and X.F.; investigation, J.W., Z.L. and Y.W.; resources, J.W., J.T. and Q.Y.; data curation, J.W. and J.L.; writing—original draft preparation, J.W.; writing—review and editing, J.W. and J.T.; visualization, J.W.; supervision, J.T.; project administration, J.T.; funding acquisition, J.T. All authors have read and agreed to the published version of the manuscript.

**Funding:** This research was funded by the National Natural Science Foundation of China (Project No. 31960330) and the Ningxia Natural Science Foundation of China (Project No. 2020AAC03112).

**Data Availability Statement:** The data presented in this study are available on reasonable request from the corresponding author.

**Conflicts of Interest:** The authors declare no conflict of interest.

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
