# Peer review of "Driving Factors and Trend Prediction for Annual Sediment Transport in the Upper and Middle Reaches of the Yellow River from 2001 to 2020"

_water, doi:10.3390/w15061107_

Round 1

Reviewer 1 Report

Dear Authors, thanks a lot for the very interesting paper, which represents a valuable contribution in using remote sensing techniques to study the driving factors and their effect in sediment unbalancing at the river basin scale. In my opinion the paper at the present stage is worth to be published.

Best regards

Author Response

Dear Reviewer,

We would like to thank you for your careful reading, helpful comments, which has significantly improved the presentation of our manuscript. The manuscript has also been double-checked, and the typos and grammar errors we found have been corrected. We believe that our responses have well addressed all concerns from the reviewers. We hope our revised manuscript can be accepted for publication.

Best wishes,

Jingjing Wu

Reviewer 2 Report

The manuscript contains a reliable and comprehensive description of the statistical analysis of the relationship between sediment transport variability and changes in vegetation, summer precipitation, soil moisture, population and land use. Sediment transport data from the five gauge stations were obtained from the Sediment Bulletin of China River. The data of factors influencing the transport of sediments from satellite monitoring from 2001-2020 were obtained from the Google Earth Engine platform. In my opinion, these data are the basis for the implementation of the assumed research goal, which was to establish a simple, feasible, and accurate model to predict the changes in sludge transport in the upper and middle reaches of the Yellow River. The developed model can effectively predict changes in sediment transport in rivers. However, this model has limitations in effectiveness due to not taking into account all the factors that have influenced changes in sediment transport in the Yellow River. Some of these limitations were pointed out by the authors in the discussion of the results. The manuscript should be published in its present form.

Author Response

Dear Reviewer,

We would like to thank you for your careful reading, helpful comments, and constructive suggestions, which has significantly improved the presentation of our manuscript. We have carefully considered all comments from the reviewer and revised our manuscript accordingly. 

The manuscript has also been double-checked, and the typos and grammar errors we found have been corrected. We believe that our responses have well addressed all concerns from the reviewers. We hope our revised manuscript can be accepted for publication.

Best wishes,

Jingjing Wu

Reviewer 3 Report

Review report: Driving factors and trend prediction for annual sediment 2 transport in the upper and middle reaches of the Yellow River 3 from 2001 to 2020

Recommendation: Major revision

              This article tries to correlate annual sediment transport (AST) in the Yellow river with 16 parameters extracted from Google Earth Engines. I think its content and the method applied are acceptable and deserve a publication in Water, after some major improvements. After reading the Introduction, I thought the Introduction was written well. Nevertheless, after reading further, more and more questions arose. Sadly, the Materials and Methods, the Discussion, and the Conclusion were disappointing. I think the authors may be statisticians, as they explained a lot about statistical analysis, but ignored important aspects of remote sensing, and real-life interconnections amongst the sediment-influencing parameters. I hope my recommendations can help improve the manuscript.

Introduction

              The Introduction was written well. It had extensive literature review on the AST in the Yellow river. It also had good review on parameters influencing the AST. The objective of the research is clear. The novelty lies in an application of Google Earth Engine to cover the vast area of the Yellow river. I do not have any comment in the Section.

Study area

              - Not every reader is Chinese. I am not a Chinese. I do not know all names of places unless explained in a map. Show all specific names of locations on the map, for example Bayan Har mountain, Bohai Sea are missing.

              - Figure 1, If possible, improve the map and its legend to an easy-to-understand numbers, such as 250-999 m, 1,000-1,499 m, 1,500 – 1,999 m………

Materials and methods

              - The authors explained well about the statistical approach used in this study, but totally ignored how they derived their 16 independent parameters in details. The authors just showed Table 2 (about the data sources), without any detailed explanations on the pre-processing of each parameter.

              - As all 16 independent parameters were analyzed to predict the AST (the dependent parameter), if you derived each independent parameter incorrectly, the whole result of your AST prediction model would be incorrect too. Please explain the pre-processing of how you derived each of the 16 parameters used in your study, so other readers can repeat your approach.

              - How did different resolution affect your derivation?  Some image resolution is 100 m, but some is 927 m.

              - How much radius was analyzed for each station? Was it 5 kilometers from each gauge station? Or 1 kilometer from the gauge station? For example, Farmland area within 1-km radius? Shrubland with 2-km radius?   Why so?  This is important because the derived value can affect your regression model.

              - It seems that the authors are statisticians who focus on the statistics, yet ignoring necessary explanations on remote sensing analysis. Your AST prediction model relied on 16 independent variables, which had to be derived separately. If you don’t present the values of each independent variable, how can we know that your analysis was reliable?   

              - The dependent variable, which is the AST was taken from the Sediment Bulletin of China River. How did it measure the sediment load in the river? What method? The frequency of measurement? The AST from the government report is your dependent variable, which is the heart of your research. If the AST was measured with a non-reliable approach, it could greatly affect your regression model.

Results

              - The authors analyzed the AST well, yet my question still remain that How did Sediment Bulletin of China River measure the sediment load in the river?

              - Values of all independent variables should be presented.

              - Section 4.3.1, Based on your regression model, If you want to predict the AST in 2021-2023, you need to predict your 5 independent variables first, don’t you?  How did you predict them?

Discussion

              - You should discuss your own results more critically. Section 5.1 is fine. Ln 390-422, this is a discussion on the change of the AST, which is the data collected by the Sediment Bulletin of China River, not by yourself.

              - You should have discussed a lot more on Sec 5.2 why each independent parameter has/does not influence the AST.

              - I am satisfied with their predicted R2 which is approximately 0.5. Section 5.3 is fine with me.

Conclusion

              - The authors should re-write the conclusion, as it now repeats the Discussion. Line 457-499 clearly say the same things in the Discussion.

Round 2

Reviewer 3 Report

Thank you for your revision. I am mostly satisfied with the responses. However, there are a few more minor suggestions. If the authors can improve their article based on theses comments, the Editor can proceed with their publication without having to send it back to me.

- Please provide units of each parameter in Table 1

- Although you responded “As most Chinese scholars rely on this data when studying river runoff and sediment, we consider it to be reliable and accurate in this study” and I understand that the Sediment Bulletin of China River is officially endorsed. However, this article is supposed to reach other readers other than Chinese. Please write and explain about the Sediment Bulletin of China River and how it has been applied by other researchers with proper references.

- I believe that the linear regression method might not be the best method to forecast the 5 independent parameters. Please explain why you did not choose to use other types of the regression.

Round 3

Reviewer 3 Report

Thank you for your revision. I am happy with the revised version. I gladly accept it in its present form. Congratulations....